# The Impact of Robotic Therapy on the Self-Perception of Upper Limb Function in Cervical Spinal Cord Injury: A Pilot Randomized Controlled Trial

**DOI:** 10.3390/ijerph19106321

**Published:** 2022-05-23

**Authors:** V. Lozano-Berrio, M. Alcobendas-Maestro, B. Polonio-López, A. Gil-Agudo, A. de la Peña-González, A. de los Reyes-Guzmán

**Affiliations:** 1Biomechanics and Technical Aids Department, National Hospital for Spinal Cord Injury, 45004 Toledo, Spain; vlozanob@sescam.jccm.es (V.L.-B.); monica.alcobendas@uclm.es (M.A.-M.); amgila@sescam.jccm.es (A.G.-A.); adlos@sescam.jccm.es (A.d.l.R.-G.); 2Technological Innovation Applied to Health Research Group (ITAS), Faculty of Health Sciences, University of Castilla-La Mancha, 45600 Talavera de la Reina, Spain; 3Occupational Therapy Unit, National Hospital for Spinal Cord Injury, 45004 Toledo, Spain; aisabelpg@sescam.jccm.es

**Keywords:** spinal cord injury, exoskeleton, robot-assisted therapy, upper limb, self-perception, activities of daily living

## Abstract

Background: The aim of the present study was to evaluate the impact of robotic therapy in patients with cervical spinal cord injury (SCI), measured on the basis of the patients’ self-perception of limited upper limb function and level of independence in activities of daily living. Methods: Twenty-six patients with cervical SCI completed the treatment after being randomly assigned to the intervention or control group. The training consisted of 40 experimental sessions 1 h in duration, ideally occurring 5 days/week for 8 weeks. In addition to the conventional daily therapy (30 min), the control group received another 30 min of conventional therapy, whereas the intervention group received 30 min of robotic therapy. Patients were evaluated by means of the Capabilities of Upper Extremity Questionnaire (CUE) and Spinal Cord Independence Measure (SCIM) clinical scales. Results: The improvement in the feeding item of SCIM was significantly higher in the intervention group than in the control group after the treatment (2.00 (0.91) vs. 1.18 (0.89), *p* = 0.03). The correlation between the CUE and SCIM scales was higher at the ending than at baseline for both groups. Conclusions: Although both groups improved, the clinical relevance related to the changes observed for both assessments was slightly higher in the intervention group than in the control group.

## 1. Introduction

Spinal cord injury (SCI) is one of the most severe injuries and produces lifelong disability with psychological, social, economic, and permanent neurologic effects [1]. It also leads to a reduction in life expectancy compared with people without SCI [2]. The National Spinal Cord Injury Statistical Center estimated that the prevalence of people with SCI living in the United States is approximately 294,000 [3], and the World Health Organization estimated that between 250,000 and 500,000 people experience an SCI each year (WHO, 2013). The annual incidence of SCI is approximately 54 cases per one million in the USA. Approximately 78% of new SCI cases are male; cervical-level injuries have a higher prevalence than those at thoracic and lumbar levels, and incomplete tetraplegic is the most frequent neurological category [3].

One of the most critical aspects of a cervical SCI is the partial or complete impairment of arm and hand function, in addition to impaired sensory and autonomic function. Moreover, the injury has a great impact on the patient’s level of independence, which causes difficulty or even the inability to perform activities of daily living (ADLs). As a consequence, patients experience a loss of quality of life [4]. Measures to prevent possible complications include early range of motion and rehabilitation [1]. Rehabilitative training itself is currently the most successful treatment to promote functional recovery following SCI [5]. Specifically, it is necessary to promote the recovery of upper limb (UL) movements and the improvement of UL functions through motivating, intense, repetitive, and rhythmic training [2,6,7].

Robotic technology has been increasingly used to assess and treat UL motor deficits by means of specific therapeutic or assistive functions [2]. These devices combine the benefits of repetitive task-oriented training with a greater intensity of practice, less dependence on therapist assistance [6], and the appropriate number of repetitions [8]. Virtual reality environments combined with robotic therapy (RT) allow us to simulate real-life activities, providing feedback to the patients [9] and increasing patients’ adherence and motivation [10].

Currently, a large variety of electromechanical devices for UL rehabilitation is commercially available. These devices can be classified according to their structure as exoskeletons and end-effectors, in addition to the increasingly used wearables [11]. The most recent studies suggest that no one type of robotic device is better or worse than any other device [12]. Moreover, because our center has Armeo^®^Spring (Hocoma AG, Switzerland), research has focused only on the use of this device. Armeo^®^Spring is a rehabilitative passive device intended for patients who have lost UL function [13]. This device is a spring-based weight compensation exoskeleton that allows virtual gaming in a three-dimensional workspace. Using weight compensation in the shoulder, elbow, and forearm and a pressure-sensitive handgrip, which allow full or partial unloading of UL, this device allows the patient to train with specific exercises in order to increase the muscle strength and range of motion in different joints, with the overall goal of improving motor function [6,14].

On the other hand, the World Health Organization and the Canadian Association of Occupational Therapy promote patient-centered practice, involving patients in the decision-making process regarding their participation in rehabilitation [14,15,16]. Gerteis et al. elaborated on seven dimensions of patient-centered care. Among them, the respect for patients’ values, preferences, strengths, capabilities, and expressed needs were included [17]. It is important to take into account the perspective of the patient and their perception of the benefits of RT on their UL capabilities to measure the impact of the injury and the outcomes after rehabilitation [18,19]. However, although the assessment of UL function and capability is carried out using clinical scales given directly to the patient, these specific methods for user involvement are rarely addressed in the literature [20].

Taking these aspects into account, studies focused on ULRT in SCI patients are still scarce and have controversial results [21] when comparing RT with conventional therapy (CT) [22]. Scientific evidence in patients with SCI suggests that the improvements observed in patients cannot be attributed to the use of the robotic device alone, as the experimental groups have not always received the same doses of therapy, and sufficient evidence of clinical effectiveness is lacking [21]. Furthermore, most studies on SCI with robotic systems for UL functional improvement have been conducted in chronic patients [21] rather than in acute–subacute patients. This suggests that further studies should be carried out in subacute patient populations.

Therefore, to our knowledge, there is no evidence from similar studies that have examined the self-perception of UL function after treatment with RT in cervical SCI patients.

Using these hypotheses as a starting point, the main objective of this study was to evaluate the impact of the robotic device Armeo^®^Spring on the perceptions that cervical SCI patients have about their UL capabilities. The secondary objective was to analyze the correlation between the self-perception of the UL capabilities and the independence level in ADLs. To achieve these objectives, the Capabilities of Upper Extremity Questionnaire (CUE) and Spinal Cord Independence Measure (SCIM) clinical scales were used. Therefore, an independent control group of subjects not undergoing Armeo Spring treatment was included, and the control and experimental groups were dose-matched.

## 2. Materials and Methods

### 2.1. Study Design and Randomization

The study was interventional with a parallel assignment model, two arms, and randomized allocation into a control or intervention group using a random number generation program and opaque envelopes following blocked randomization. For each block of four participants, two were allocated to each group in the trial. The random allocation sequence was generated by a researcher who was external to this study. Then, the participants were enrolled and assigned to interventions for clinical staff. The examiners were unaware of the experimental group assignment. The clinical trial ended when the study reached the scheduled date of closure. The study was registered at ClinicalTrials.gov (NCT0383873). The CONSORT flow chart is shown in Figure 1.

### 2.2. Participants

A total of 32 patients with subacute cervical SCI were recruited between April 2016 and April 2019 from the inpatient population at Hospital Nacional de Parapléjicos in Toledo (Spain). During the screening process, all participants were interviewed and examined by a study researcher with more than 5 years of clinical experience in neurological rehabilitation. All the clinical and neurological examinations were made by the same clinical professionals. The inclusion criteria were as follows: a cervical SCI with a motor level between C4 and C8 with a classification on the ASIA Impairment Scale (AIS) between A and D, as evaluated by clinical staff; a traumatic or non-progressive medical etiology of the injury; less than 6 months of injury evolution (subacute); age between 16 and 75 years; and achievement of a seated posture. Patients also needed to be fully informed and sign the corresponding informed consent to participate in the study. The exclusion criteria were unstable orthopedic injuries, such as unconsolidated fractures, or unstable osteosynthesis systems in UL; skin lesions and/or pressure ulcers in the exoskeleton placement area; joint stiffness and/or severe spasticity; broncho-pneumopathy and/or severe heart disease that required monitoring during exercise; and visual problems and/or cognitive impairment. Patients who did not sign the corresponding informed consent were also excluded. Three patients were excluded because they did not meet the inclusion criteria, and another declined to participate.

### 2.3. Ethics

Prior to inclusion, both verbal and written informed consent were obtained from all the participants. The study was conducted in accordance with the Declaration of Helsinki, and the protocol was approved by the Ethics Committee of Clinical Research of the Complejo Hospitalario de Toledo (approval number: 14/06/2016 Nº85).

### 2.4. Interventions

The study was organized into two arms: the experimental arm, composed of the intervention group, and the active comparator arm, composed of the control group. The treatment was unilateral. The choice of the UL treated was made according to the rehabilitation objectives and by consulting with each patient.

All the patients received 30 min of CT daily based on UL functional and ADL treatment. In addition, patients in the control group received another 30 min of CT, whereas patients in the intervention group received 30 min of ULRT. Thus, all the patients were scheduled for five 1 h sessions per week for 8 weeks to complete a total of 40 sessions. A maximum of 10 weeks was permitted to complete the treatment. Each ULRT session by means of the Armeo^®^Spring device (Figure 2) was divided into two parts: (1) normalized games for 15 min within Armeo software and (2) training of the ADL of drinking for another 15 min. The ADL of drinking using a virtual application was designed and developed through the collaboration that our center has maintained with the Technological Centre VICOMTECH. In the game, the patient, instrumented with the Armeo Spring device, has to reach and lift a glass on the table, perform a UL proximal movement to the mouth, simulate a swallow, perform the UL distal movement to the table, release the glass, and return to the initial point.

All the patients were evaluated using a set of clinical scales at two different time points: at baseline (before starting the treatment) and at the end of all the experimental sessions (during week 10). The Capabilities of Upper Extremity Questionnaire (CUE) [23] and the Spinal Cord Independence Measure (SCIM) total and self-care subscale (SCIM-SC) were applied in the present study.

### 2.5. Capabilities of Upper Extremity Questionnaire (CUE)

The CUE scale was designed to assess the amount of difficulty experienced in performing specific actions with one or both arms in cervical SCI [21] and offers a self-reported measure of UL functional capacity and limitations for people with tetraplegia [24]. The CUE instrument is a questionnaire with 32 items developed to evaluate the difficulty in performing determined UL tasks according to the patient’s perception. Responses to each item are given on a 7-point scale representing self-perceived difficulty in performing the task, with 1 representing “unable to perform” and 7 representing “can perform without difficulty”. Thus, the minimum total score is 32 and the maximum is 224 [23].

The following variables were used to express the results: the CUE scores for the right and left ULs; the CUE score in bilateral actions; and the total CUE score.

### 2.6. Independence in Activities of Daily Living (SCIM)

The SCIM is a disability scale developed to specifically address the ability of SCI patients to perform basic ADLs independently. It assesses three areas: self-care (feeding, grooming, bathing, and dressing), respiration and sphincter management, and mobility (bed and transfers and indoor/outdoor). Total SCIM scores range from 0 to 100; higher scores are indicative of more independence, and the subscale scores are self-care (0–20), respiration, and sphincter management and mobility (0–40) [25,26]. The SCIM total and self-care subscale (SCIM-SC) were used as comparator scores. The SCIM-SC includes items specific to UL use and was, therefore, ideal for comparisons with the CUE [27,28] and assessing changes in areas of self-care, including feeding, bathing, dressing, and grooming.

### 2.7. Data Analysis

Statistical analysis was performed using SPSS software (version 17.0 for Windows). Descriptive statistics were performed for all variables measured, and the results are expressed as means and standard deviations with a CI of 95%.

The Kolmogorov–Smirnov test was applied to verify the normality of baseline data. Therefore, parametric tests were applied to both groups. Results are expressed as means and standard deviations. At baseline, the homogeneity of both groups was proven using the Levene test. The independent *t*-test was used to compare the score changes between both groups after the treatment. To analyze the differences between the baseline and the end of treatment, a paired-samples *t*-test was used. The effect size (ƞ^2^) was computed following the guidelines proposed by Cohen, and this value was interpreted as follows: 0.01 = small effect, 0.06 = moderate effect, and 0.14 = large effect [29]. To analyze the correlation, the Pearson correlation coefficient was used. The values obtained were interpreted as good to excellent (values greater than 0.75), moderate to good (values between 0.50 and 0.75), or fair (values between 0.25–0.50) [26].

## 3. Results

Twenty-eight participants were enrolled and randomly allocated to either the control group or the intervention group. Two patients dropped out before ending the rehabilitation treatment. One received UL treatment by means of another device (Hand Tutor). The other patient was discharged from the hospital before the end of the treatment. Finally, 26 patients were analyzed, with 13 participants in each group. The demographic and functional characteristics of the 26 participants are shown in Table 1, which shows the homogeneity of both groups at baseline. Therefore, both groups were matched with respect to the cervical metameric level and severity of injuries affecting the subjects.

### 3.1. Outcomes between Baseline and Ending Treatment Assessments

Regarding the CUE clinical scale, statistically significant improvements were found in all the variables analyzed for both groups, except for actions in which both ULs were involved (Table 2). Furthermore, regarding the effect size, a large effect was observed for all the items in both groups, with exception of tasks involving both ULs for the intervention group (ɳ^2^ = 0.09). Comparing the effect size between the two groups, it can be seen that the total score on the CUE scale in the intervention group was slightly higher than that in the control group (0.52 vs. 0.47, respectively), and a greater effect size was observed for the right UL in the intervention group than for the right UL in the control group (0.55 vs. 0.41, respectively). By contrast, a greater effect size was observed in the item related to both extremities in the control group than in the intervention group (0.26 vs. 0.09, respectively).

With respect to the SCIM clinical scale, significant statistically improvements were observed for the total score, the self-care area, and the bathing item for both groups analyzed (Table 2). Moreover, for these variables, the changes observed between the two assessments were clinically relevant in both groups with a large effect size (ɳ^2^ > 0.14). Although this high clinical relevance was observed in the dressing item, statistically significant differences were observed only for the control group (*p* < 0.05). Regarding the feeding item, the improvements observed between both assessments were statistically significant in the intervention group (*p* = 0.009). However, the changes observed were clinically relevant for both groups (ɳ^2^ > 0.14), although the effect size was higher in the intervention group (ɳ^2^ = 0.44) than in the control (ɳ^2^ = 0.20).

### 3.2. Changes between Groups after the Treatment

The differences between the two groups were analyzed at the end of the assessment (Table 2).

Regarding the CUE clinical scale, no statistical differences were found between the two groups, and the results observed at the end of the study were greater in the control group than in the intervention group.

Regarding the SCIM scale, the level of a patient’s independence for the feeding item was significantly higher in the intervention group than in the control group (2.00 (0.91) vs. 1.18 (0.89), *p* = 0.03). No statistically significant differences were found in the other items or in the total SCIM score, but for these variables, the results observed were slightly higher in the control group than in the intervention group.

### 3.3. Correlation between CUE and SCIM Clinical Scales

The correlation between the CUE and SCIM clinical scales was analyzed, and the results are shown in Table 3. At baseline, the correlation between the CUE total score and the SCIM scale was statistically significant for only three items in both experimental groups. However, at the end of the treatment (week 10), this correlation was statistically significant for the total SCIM, the SCIM-SC, and each item within the self-care area. At the end of treatment, the correlation between the self-care items was high for both experimental groups (0.837 in the control group and 0.821 in the intervention group, *p* < 0.01). Specifically, on the feeding item, the correlation obtained was high for the intervention group (0.835, *p* < 0.01) and moderate for the control group (0.666, *p* < 0.05).

The correlation analysis was particularized to cases in which the treated UL was the dominant one. A total of eight patients within each experimental group fulfilled this condition. The results are shown in Table 4. In the control group, the results obtained for the whole sample were maintained. However, in the intervention group, the differences detected between the evaluations at baseline and at the end of the treatment were greater (Table 4) than those observed for the whole sample (Table 3). Thus, for this group, statistically significant correlations were obtained at the end of the assessment between the CUE and the self-care area of the SCIM scale, specifically, the items on the feeding, bathing, and dressing of the upper body and grooming (Table 4).

Specifically, the highest correlation obtained in this particularized analysis was 0.848 (*p* < 0.01) for the food item, while in the previous analysis, this same correlation was 0.835 (*p* < 0.01). It is worth highlighting the increase in the correlation obtained for the total score on the SCIM scale in both groups (0.932 (*p* < 0.01) for the control group and 0.706 (*p* < 0.01) for the intervention group).

## 4. Discussion

In the present study, the self-perception of patients with cervical spinal cord injury regarding difficulties and limitations in UL function was analyzed by means of the CUE scale. In addition, the correlation between this scale and the level of independence in ADLs assessed by means of the SCIM scale has been analyzed. These aspects were assessed in two different therapeutic interventions, and the impact of RT can be evaluated in terms of the variables described.

Over the last two decades, the use of robot-assisted training has become more and more popular in the field of rehabilitation. As result, efficacy measurements on UL function, independence in ADLs, and movement quality seem to be positive, but with sparse and ambiguous results [30]. The effectiveness of RT over CT is arguable, and the best therapy strategy is still not clear [31]. Mehrholz et al. published a systematic review and meta-analysis evaluating different electro-mechanical devices for improving ADLs and hand–arm function after stroke and concluded that robotic-assisted arm training was comparable to CT. Moreover, the results suggest that no one type of robotic device is better or worse than any other device [11].

Studies focused on cervical SCI are still scarce and have controversial results, mainly due to the fact that the experimental groups have not always received the same doses of therapy, and they lack sufficient evidence of clinical effectiveness [21]. Thus, Zariffa et al. concluded that, while robotic-assisted training is feasible in an inpatient setting, it offers few functional benefits when compared with CT [8]. Sørensen et al. performed a single-subject study of robotic training using Armeo^®^Spring, which included four participants and obtained similar outcomes to those obtained by Zariffa et al. [32].

Most of the studies that have evaluated the effect of robotic training on arm and hand motor functions have not included the CUE clinical scale. Thus, Oleson and Marino [24] determined that the revised version of the CUE scale, with only five items, retains the responsiveness of the original version. In a study published by Kalsi-Ryan et al. [33], the CUE scale was used as a measurement to establish the relationship between the impairment and the self-perceived UL function. Prasad et al. [34] used the CUE scale to assess the functional ability of the treated hand to determine the effectiveness of virtual reality therapy over the UL function in patients with SCI. Sinnott et al. [19] included this scale in their narrative review of literature as a measurement tool for UL reconstructive surgery for cervical SCI. Kalsi-Ryan et al. [35] evaluated the sensorimotor impairment and function of the arm and hand by means of the CUE scale, among other measurements, to define the sensory, motor, and prehension recovery profiles of the UL and hand. Sledziewski et al. [36] examined the use of the Reo Go^®^ robotic device to treat UL dysfunction in cervical SCI using the CUE scale to assess perceived changes in the right UL function in a case report.

Regarding the SCIM clinical scale, Sørensen et al. [32] did not obtain clinically significant changes in the independence level in ADLs and concluded that the increase obtained in SCIM was not likely to be related to the robotic intervention. Francisco et al. [2] obtained similar results, and no statistically significant differences were found after the treatment regarding the independence level in ADLs.

In the study of Jung et al. [22], two robots, Armeo^®^Power and Amadeo^®^, were combined to cover the training of the proximal and distal UL. In the SCIM scale, the intervention group showed significant improvements in three items (bathing the upper body, dressing the upper body, and grooming), whereas the control group showed improvements in only one item (dressing the lower body). However, both groups showed a significant increase in the total score on the SCIM scale. However, to our knowledge, there is no evidence of similar studies with the aim of applying the CUE as an evaluation tool during UL training by means of a robotic device, especially Armeo^®^Spring. For this reason, the results obtained in the present research cannot be compared with those of other studies. However, the results obtained were interesting. In our study, a significant improvement in the perception of UL function after treatment was received was observed in both groups. In the intervention group, the clinical relevance observed between baseline and the end of treatment was greater in the scoring related to the right UL. This finding could be because the treatment by means of Armeo Spring was unilateral, and in most cases, the UL treated was the right arm. However, in the control group, the clinical relevance was greater for the total scoring of the CUE scale, probably due to the fact that the entire time of treatment was divided between the two arms.

Regarding the SCIM scale, UL functional improvements were found in both groups with statistical significance. For the SCIM-SC items, in the control group, there were statistically significant changes in bathing (upper and lower body) and dressing (upper and lower body), activities included in the CT program. In the intervention group, statistically significant changes were found in feeding, bathing (upper and lower), and grooming items, probably due to the fact that the drinking ADL was included within the therapy by means of Armeo^®^Spring. In this regard, Jung et al. [22] concluded that the improvement noted in ADLs evaluated by SCIM could be attributed to the improvement in the strength of the proximal muscles by the use of the robotic device.

In the comparison between the two experimental groups in the evaluation at the end of the study, statistically significant differences were only found for the feeding item, with the independence level higher for the intervention group than for the control group. This fact could be because patients within the intervention group worked on the movement pattern related to the ADL of drinking from a glass by means of the robotic device Armeo^®^Spring.

With respect to the correlation analyzed between both scales, a moderate to excellent correlation was observed between the CUE and the total SCIM and SCIM-SC in both groups. This correlation was higher when the dominant UL received the treatment, highlighting the importance of dominance in studies related to UL analysis.

Some limitations were identified in this study. Firstly, the study was performed in a single center, and it was not possible to compare results with those obtained in other centers. Another important limitation is not having an additional follow-up evaluation of the participants approximately two months after the end of the study. This is because most of the patients were discharged at a date very close to the end of the study. This evaluation would have made it possible to check whether the greater relationship between the CUE and SCIM scales at the end of the study was maintained after the study had ended. Despite these limitations, this study also has strengths that are worth highlighting. Firstly, the study was carried out with the coordinated collaboration of three different departments, including physicians, occupational therapists, and engineers. The second strength is related to the methodology being designed to carry out the study in terms of therapeutic doses. In this study, both experimental groups received the same amount of therapy. All participants received their usual CT dose of 30 min per session. Additionally, participants in the control group received another 30 min of CT, while participants in the intervention group received 30 min of RT per session. The formation of an independent control group was the main difference in methodology with respect to the first published full report on RT in spinal cord injury patients [8].

## 5. Conclusions

The results of this research were interesting. Although it was observed that both experimental groups improved in terms of the variables analyzed, for many variables, it was found that the changes observed between the initial and final evaluations were more clinically relevant in the intervention group. Likewise, the relationship between the clinical scales CUE and SCIM, both specific to spinal cord injury, underwent a positive adaptation throughout the treatment and was higher in the evaluation at the end of the study.

In conclusion, it is worthwhile to continue working with this same methodology to increase the sample and confirm these findings. Similarly, the next step is to use this same study design to perform a complete analysis of the effectiveness of robotic therapy in patients with spinal cord injury.

## Figures and Tables

**Figure 1 ijerph-19-06321-f001:**
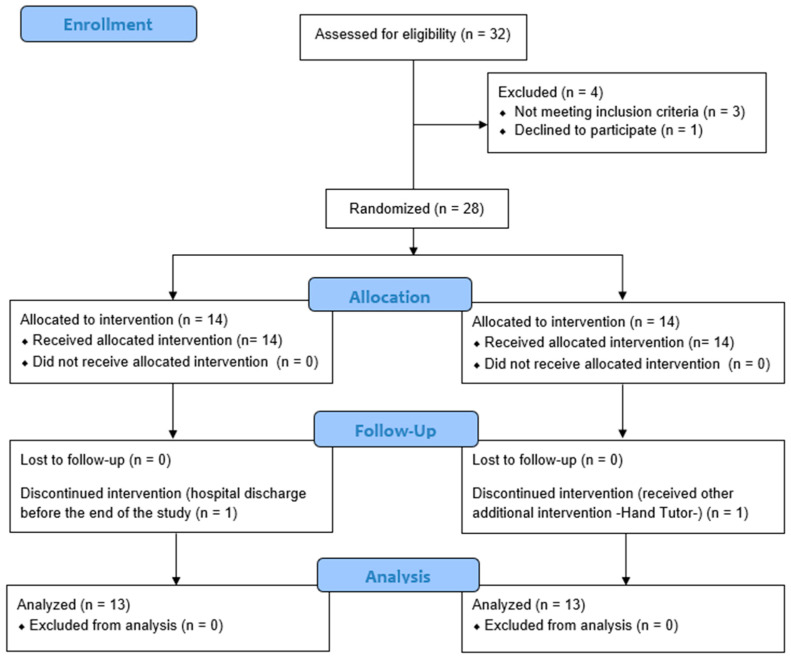
CONSORT flow chart.

**Figure 2 ijerph-19-06321-f002:**
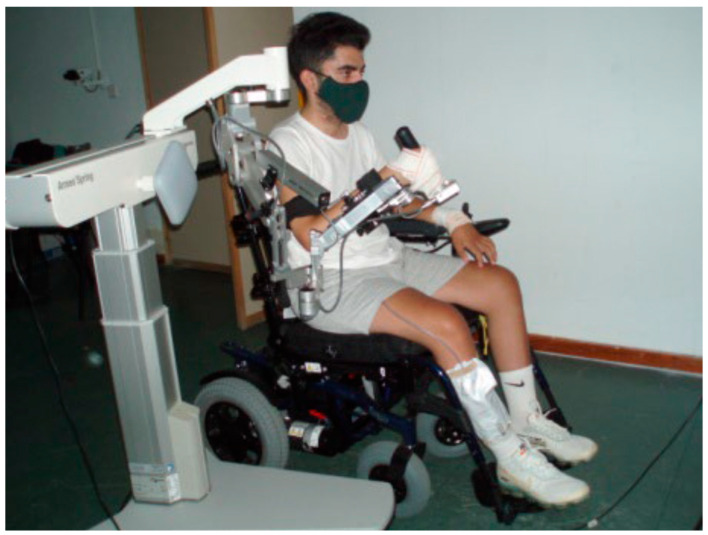
Patient during an experimental session with Armeo^®^Spring device.

**Table 1 ijerph-19-06321-t001:** Demographic and functional characteristics of the sample analyzed.

Variables	Sample Analyzed		
	Control Group (*n* = 13)	Intervention Group (*n* = 13)	Levene Test
Sex (Male) *	10.00 (76.90)	8.00 (61.52)	F = 2.623, *p* = 0.118
Age (years) ^+^	46.81 (16.30)	39.92 (16.52)	F = 0.253, *p* = 0.620
Injury level *			
C2	-	1.00 (7.69)	
C3	1.00 (7.69)	1.00 (7.69)	
C4	4.00 (30.76)	3.00 (23.07)	F = 0.176, *p* = 0.678
C5	6.00 (46.14)	6.00 (46.14)	
C6	-	1.00 (7.69)	
C7	2.00 (15.38)	1.00 (7.69)	
AIS classification *			
A	3.00 (23.07)	3.00 (23.07)	
B	1.00 (7.69)	3.00 (23.07)	F = 0.189, *p* = 0.667
C	2.00 (15.38)	1.00 (7.69)	
D	7.00 (53.83)	6.00 (46.14)	
Time since injury (months) ^+^	4.29 (1.37)	3.86 (1.66)	F = 1.578, *p* = 0.222
Dominant arm (right) *	12.00 (92.31)	12.00 (92.31)	F = 0.000, *p* = 1.000
Treated arm (right) *	10.00 (84.62)	9.00 (76.93)	F = 0.729, *p* = 0.402
Dominant and treated arm (right) *	8.00 (61.52)	8.00 (61.52)	F = 0.610, *p* = 0.443
UER/UEL (0–25) ^+^ (dominant arm)	15.63 (4.43)	14.61 (5.88)	F = 0.114, *p* = 0.738
UER/UEL (0–25) ^+^ (treated arm)	16.27 (4.90)	13.69 (5.73)	F = 0.516, *p* = 0.479

^+^ Continuous variables are expressed as means and standard deviations; * categorical variables are expressed as frequencies and percentages.

**Table 2 ijerph-19-06321-t002:** Clinical scales at baseline and at the end of the UL treatment for the control and intervention groups.

SCI Patients (*n* = 26)											
	Control Group (*n* = 13)					Intervention Group (*n* = 13)					p_2_
	At Baseline	At Ending				At Baseline	At Ending				
	Mean (SD)	Mean (SD)	t	p_1_	ᶯ ^2^	Mean (SD)	Mean (SD)	t	p_1_	ᶯ^2^	
**CUE scale**											
Total score	134.41(34.06)	153.41 (41.57)	−2.971	0.013	0.47 **	117.23 (49.43)	145.69 (53.33)	−3.598	0.004	0.52 **	0.724
Right UL	61.75 (18.93)	70.16 (24.01)	−2.763	0.018	0.41 **	51.00 (21.90)	67.53 (23.62)	−3.862	0.002	0.55 **	0.978
Left UL	66.25 (24.70)	76.00 (25.28)	−2.603	0.025	0.38 **	59.61 (30.38)	70.76 (29.08)	−2.697	0.019	0.38 **	0.724
Both UL	6.41 (3.96)	7.25 (4.28)	−1.968	0.075	0.26 **	6.61 (4.11)	7.38 (4.36)	−1.115	0.287	0.09 *	0.978
**SCIM-III scale**											
Total score	33.00 (20.13)	50.00 (24.51)	−5.079	0.000	0.68 **	31.00 (17.85)	47.00 (24.50)	−3.586	0.004	0.52 **	0.663
Self-care	4.30 (4.85)	8.84 (6.38)	−3.255	0.007	0.47 **	3.53 (3.82)	8.07 (6.84)	−3.153	0.008	0.45 **	0.681
Feeding	1.30 (1.18)	1.18 (0.89)	−1.723	0.110	0.20 **	1.15 (1.21)	2.00 (0.91)	−3.091	0.009	0.44 **	0.030
Bathing—upper	0.30 (0.75)	0.92 (1.03)	−2.889	0.014	0.41 **	0.00 (0.00)	0.92 (1.11)	−2.984	0.011	0.43 **	0.934
Bathing—lower	0.23 (0.59)	0.84 (0.89)	−2.889	0.014	0.41 **	0.00 (0.00)	0.69 (0.94)	−2.635	0.022	0.37 **	0.609
Dressing—upper	0.61 (1.19)	1.84 (1.72)	−2.792	0.016	0.39 **	0.84 (1.46)	1.61 (1.85)	−2.132	0.054	0.27 **	0.628
Dressing—lower	0.30 (0.75)	1.38 (1.66)	−2.592	0.024	0.36 **	0.30 (0.85)	1.00 (1.63)	−1.671	0.121	0.19 **	0.374
Grooming	1.53 (1.26)	2.00 (1.15)	−1.196	0.255	0.11 *	1.23 (1.01)	1.84 (1.14)	−2.309	0.040	0.31 **	0.706

Values are presented as means and standard deviations. Effect size: * moderate effect (>0.06), ** large effect (>0.14). p_1_^:^ Intragroup comparison (comparison of baseline and ending scores with paired-sample *t*-test), p_2_: intergroup comparison (comparison of differences in baseline–ending scores with the independent *t*-test).

**Table 3 ijerph-19-06321-t003:** Correlation coefficient between the total CUE score and SCIM clinical scale at baseline and ending for both experimental groups.

	SCI Patients (*n* = 26)			
	Control Group (*n* = 13)		Intervention Group (*n* = 13)	
	At Baseline	At Ending	At Baseline	At Ending
Total SCIM	0.323	0.711 **	0.481	0.678 *
Self-Care SCIM	0.611 *	0.837 **	0.613 *	0.821 **
Feeding	0.756 **	0.666 *	0.478	0.835 **
Bathing—upper	0.292	0.731 **	0.000	0.767 *
Bathing—lower	0.255	0.732 **	0.000	0.684 **
Dressing—upper	0.431	0.656 *	0.611 *	0.777 **
Dressing—lower	0.292	0.716 **	0.116	0.636 *
Grooming	0.763 **	0.818 **	0.760 **	0.770 **

* (*p* < 0.05); ** (*p* < 0.01).

**Table 4 ijerph-19-06321-t004:** Correlation between CUE (total scoring) and SCIM clinical scale when the dominant UL was the treated UL for both groups analyzed.

	Control Group (*n* = 8)		Intervention Group (*n* = 8)	
	Pre	Post	Pre	Post
Total SCIM	0.202	0.932 **	0.504	0.706 *
Self-Care SCIM	0.707 *	0.852 **	0.591	0.825 *
Feeding	0.781 *	0.735 *	0.577	0.848 **
Bathing—upper	0.000	0.711 *	0.000	0.806 *
Bathing—lower	0.000	0.711 *	0.000	0.632
Dressing—upper	0.310	0.711 *	0.618	0.810 *
Dressing—lower	0.000	0.747 *	−0.055	0.581
Grooming	0.754 *	0.888 **	0.847 **	0.720 *

* (*p* < 0.05); ** (*p* < 0.01).

## Data Availability

The study was registered in Clinical Trials, NCT04383873. Data are available at https://clinicaltrials.gov/ct2/show/study/NCT04383873 (accessed on 12 May 2020).

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
