# Peer review of "The Impact of Robotic Therapy on the Self-Perception of Upper Limb Function in Cervical Spinal Cord Injury: A Pilot Randomized Controlled Trial"

_ijerph, 2022, doi:10.3390/ijerph19106321_

Round 1
Reviewer 1 Report
The manuscript titled “The impact of robotic therapy on the self-perception of upper limb function in cervical spinal cord injury: A pilot randomized controlled trial” is interesting. The writing is difficult to comprehend due to the long and complex sentence structure and various spelling and grammar mistakes. This starts as early as the first sentence in the introduction: “The spinal cord injury (SCI) is one of the most severe injuries that causes lifelong disability, with psychological, social, economic, and permanent neurologic effects and whose, the life expectancy is less than persons without SCI.” I believe that “The” in the beginning of the sentence is not necessary. Moreover, “whose” should be replaced with something along the lines of “for those that are affected”. Lastly the sentence should be divided into 2-3 smaller sentences. Indeed, the authors continue to use such long sentences throughout the manuscript some of which span as much as 5 lines and are composed of various embedded sentences (example: Line 62-66). I strongly suggest changing the phrasing of many passages throughout the manuscript. Please find below a summary of major and minor comments. Some minor comments only contain a reference to a specific set of lines in the manuscript. In this case, the phrasing of this part of the manuscript was either difficult to understand and/or contains spelling and/or grammar mistakes.
Major Comment
Effect size is not clearly defined. I was unable to find the guidelines proposed by Cohen on the eta-squared value. I am not sure why the effect-size is even important when the P-values of the t-test are given. Furthermore, the authors fail to explain why the values of the effect-size were interpreted in the presented manner. Further discussion is needed.
Major Comment
The authors discuss the effect of the CUE scale on Both UL but the P-Value is larger than 0.05 and therefore not significant. Further discussion is needed as to why the conclusion is not just that no significant improvement was observed.
Major Comment
The authors compare the robotic-assistance group with the control group, yet no statistical test was done comparing these two groups.
Major Comment:
Cue Scale is reported for UL but if I understand correctly only one side was treated. Yet improvements happened on both sides independently. Shouldn´t the CUE scale be divided into Treated UL and untreated UL rather than Right and Left UL. Also how come both right and left UL improve but not both together?
Major Comment:
From what I understand about the robotic assistance, a task relating closely to feeding, upper bathing and grooming was trained. The authors conclude that feeding, upper bathing, and grooming improved more in intervention group than in the control group. However, it remains unclear whether that was due to the robotic-assistance or the addition of similar movements in the training protocol. The manuscript would benefit greatly from more information on how the conventional therapy was conducted and a discussion on what is the actual cause of the improvements.
Minor Comment:
Some abbreviations – such as CUE or SCIM-III - are not explained in the abstract. Furthermore, these abbreviations are explained multiple times throughout the text (for example SCIM-III in line 180 and 196).
Minor Comment:
Closing Brackets after SCIM-III but no opening brackets in line 27.
Minor Comment:
Line 38-40
Minor Comment:
No citation in line 43-44
Minor Comment:
Line 44-47
Minor Comment:
Line 48-52
Minor Comment:
Line 52-53
Minor Comment:
Line 53-57
Minor Comment:
Line 62-66
Minor Comment:
Double spacing in line 68
Minor Comment:
Line 71-74
Minor Comment:
Line 77-81
Minor Comment:
Line 82-84
Minor Comment:
Line 86-90
Minor Comment:
Line 96-101
Minor Comment:
Line 102-106
Minor Comment:
Line 116-121
Minor Comment:
I think there are some mistakes in Table 1 for example: 8 out of 13 patients had a Dominant and treated arm (right) which equates to 61.52% but the value is 76.93.
Minor Comment:
Line 154-156
Minor Comment:
Would benefit from a sketch on how the robotic therapy works.
Minor Comment:
Line 173-176
Minor Comment:
Line 177-181
Minor Comment:
Line 198-200
Minor Comment:
Line 220-223 is a repetition of methods.
Minor Comment:
The abbreviated from of the effect size (eta-squared) is often missing and instead only a double spacing with a 2 is visible (example: Line 226).
Minor Comment:
Line 226-230
Minor Comment:
Line 267-273
Minor Comment:
Line 287-289
Minor Comment:
Line 293-294
Minor Comment:
Line 302-304
Minor Comment:
Line 306-312
Minor Comment:
Line 313-315
Minor Comment:
Line 324-327
Minor Comment:
Line 333-337
Minor Comment:
Line 229-341
Minor Comment:
Please elaborate on the strengths of this study (line 341-343).
Minor Comment:
Line 345-352
Reviewer 2 Report
L28 how many sessions per week?
Explain the acronyms, insert a mean difference or a significance index in the results
L32 I recommend less cryptic conclusions
L35 add: end-effector, robot-assisted therapy
L67 missing reference..
Ref: Qassim HM, Wan Hasan WZ. A Review on Upper Limb Rehabilitation Robots. Applied Sciences. 2020; 10(19):6976. https://doi.org/10.3390/app10196976
L69 It is not appropriate to call it passive, but end-effector. For this reason I suggest arguing the concepts of Robot Assisted Therapy of the upper limb with 3 devices: End-effector, Exoskeleton and Wearable. You use an end-effector instrumentation that guarantees less functionality, but is much more reliable in patients with SCI. I can suggest adding a reference regarding the comparison of devices:
Ref: Moggio L, de Sire A, Marotta N, Demeco A, Ammendolia A. Exoskeleton versus end-effector robot-assisted therapy for finger-hand motor recovery in stroke survivors: systematic review and meta-analysis [published online ahead of print, 2021 Aug 21]. Top Stroke Rehabil. 2021;1-12. doi: https://doi.org/10.1080/10749357.2021.1967657
102 robotic end-effector device
110 … 28 is a result of a selection.. move it in appropriate section. Please, describe only the eligibility in this section
128 Remove the allocation results, but instead claim that you used an allocation with a 1: 1 ratio
135 Results
146 Results
222 Insert Flow Chart.. then the characteristic of enrolled participants
341 The strengths of the study are inappropriate, a limitation being the failure to use a widespread scale such as the DASH
Author Response
Please, see the attachment.
